# Preparation, Characterization, and Evaluation of Liposomes Containing Oridonin from *Rabdosia rubescens*

**DOI:** 10.3390/molecules27030860

**Published:** 2022-01-27

**Authors:** Yinyue Wang, Mai Wang, Feier Lin, Xinyan Zhang, Yongming Zhao, Chunyan Guo, Jin Wang

**Affiliations:** 1Department of Pharmacy, Hebei North University, Zhangjiakou 075000, China; yinyue8709@163.com (Y.W.); maiwang5030@163.com (M.W.); PhilLin2022@163.com (F.L.); zxy98112369@163.com (X.Z.); guochy0311@163.com (C.G.); 2Hebei Key Laboratory of Neuropharmacology, Zhangjiakou 075000, China

**Keywords:** *Rabdosia rubescens*, oridonin, ultrasound-assisted extraction, liposomes

## Abstract

Due to the remarkable anti-tumor activities of oridonin (Ori), research on *Rabdosia rubescens* has attracted more and more attention in the pharmaceutical field. The purpose of this study was to extract Ori from *R. rubescens* by ultrasound-assisted extraction (UAE) and prepare Ori liposomes as a novel delivery system to improve the bioavailability and biocompatibility. Response surface methodology (RSM), namely Box-Behnken design (BBD), was applied to optimize extraction conditions, formulation, and preparation process. The results demonstrated that the optimal extraction conditions were an ethanol concentration of 75.9%, an extraction time of 35.7 min, and a solid/liquid ratio of 1:32.6. Under these optimal conditions, the extraction yield of Ori was 4.23 mg/g, which was well matched with the predicted value (4.28 mg/g). The optimal preparation conditions of Ori liposomes by RSM, with an ultrasonic time of 41.1 min, a soybean phospholipids/drug ratio of 9.6 g/g, and a water bath temperature of 53.4 °C, had higher encapsulation efficiency (84.1%). The characterization studies indicated that Ori liposomes had well-dispersible spherical shapes and uniform sizes with a particle size of 137.7 nm, a polydispersity index (PDI) of 0.216, and zeta potential of −24.0 mV. In addition, Ori liposomes presented better activity than free Ori. Therefore, the results indicated that Ori liposomes could enhance the bioactivity of Ori, being proposed as a promising vehicle for drug delivery.

## 1. Introduction

*Rabdosia rubescens* Hara, which is known as Donglingcao, also named Poxuedan, is mainly distributed in the dry slopes and along the streams in the northern provinces of China. People use *R. rubescens* for gum disease, prostate, cancer, and other conditions. Oridonin (Ori), an *ent*-kaurane diterpenoid with formula C_20_H_28_O_6_, is the main active ingredient isolated from the leaves of *R. rubescens* [1,2]. According to the literature review, Ori has antiviral, anti-tumor [2,3,4,5,6], anti-inflammation [7], and anti-bacterial properties as well as scavenging activity against oxygen free radicals [8]. It has also been confirmed that Ori effectively inhibits the proliferation of more than 20 human cancer cell lines [9,10], including those from the breast [11,12], prostate [13], lung, and liver [14]. However, poor solubility, low bioavailability, and instability limit its application as an effective therapeutic agent. Therefore, the development of new drug carriers to overcome these problems has become a hot spot in scientific research.

Liposomes, spherical shape vesicles composed of one or more phospholipid bilayer membranes, were first used as a novel drug carrier in 1971 [15]. The internal cavity of the liposomes allows the encapsulation of polar drug molecules. Amphiphilic and lipophilic molecules can be dissolved in the phospholipid bilayer. Due to being nontoxic, biodegradable, and biocompatible with the cell membrane, and with multiple functions including toxicity reduction, solubility and stability improvement of the encapsulated drugs, liposomes have been widely used as a drug carrier [16,17]. Up to now, Food and Drug Administration (FDA) has approved more than 10 liposomal drugs, and many more preparations are undergoing clinical trials. Several research institutions and researchers have reported using liposomes as drug carriers for anti-cancer, anti-fungal, antibiotic, anti-inflammatory drugs and gene medicines [18,19,20]. 

Response surface methodology (RSM), introduced by Box and Wilson, is a collection of mathematical and statistical techniques used to evaluate the interactions between several independent variables and responses [21]. The main purpose of RSM is to maximize production yield by optimizing the response surface that is influenced by various process variables. By careful design of experiments, fewer experimental trails can be used to optimize interactions between multiple process variables. Therefore, RSM has been widely used for developing, improving, and optimizing processes [22,23]. Box-Behnken design (BBD) is a conventional method for experimental design which was applied in this study to optimize the extraction conditions and preparation conditions of Ori liposomes.

In this study, Ori was efficiently extracted from *R. rubescens* by using ultrasound-assisted extraction (UAE) and encapsulated with liposomes to improve its bioavailability. Moreover, physicochemical characteristics of the Ori liposomes as well as anti-proliferation activity of breast cancer cells (MCF-7) were investigated. The aim of this study is to optimize the preparation conditions of Ori liposomes and assess anti-breast cancer cell activity in vitro, which provides the theoretical basis for the further exploration, development, and utilization of Ori from *R. rubescens*.

## 2. Materials and Methods

### 2.1. Materials and Reagents

Whole grass of *R. rubescens* was purchased from the An-Guo herbal medicine market (Hebei, China). The dried grass was ground into powder and sieved with 80-mesh screens, and then stored in a desiccator at room temperature for further experiments. The Ori reference (>98%) and acetonitrile of chromatographic grade were obtained from Adamas-beta (Shanghai, China). Analytical-grade ethanol was used for extraction. Soybean phospholipids (SY-SO-200801) and cholesterol (B80859) were purchased from Shanghai Aiweituo Pharmaceutical Technology Co., Ltd. (Shanghai, China).

### 2.2. Extraction and Purification of Ori from R. rubescens

#### 2.2.1. Ultrasound-Assisted Extraction 

UAE procedure was performed by mixing 5 g of *R. rubescens* powdered sample with 150 mL ethanol at different concentrations in a flask for 30 min using a KQ-400KDE ultrasonic water bath at a frequency of 40 KHz with power of 100 W (Kunshan, China). Both temperature and time can be easily adjusted through the control panel. The extraction was carried out in a 30 °C water bath for different times. After extraction, the supernatant was filtered through a 0.22 μm cellulose filter and the Ori content was determined by ultra high-performance liquid chromatography (UPLC) [24]. The extraction yield (%) was calculated as follows in Equation (1):(1)$$\mathrm{yield}\;\left(w/w\right)=\frac{\mathrm{Ori}\;\mathrm{weight}\;(\mathrm{mg})}{\mathrm{Powdered}\;\mathrm{sample}\;\mathrm{weight}\;(g)}$$

#### 2.2.2. Experimental Design

Based on the results of preliminary experiments, three-variable-three-level BBD was employed through RSM to optimize the extraction process of Ori from *R. rubescens*. Three experimental factors, including ethanol concentration (40–80%), extraction time (30–50 min), and liquid/solid ratio (20:1–40:1), were chosen as independent variables for the extraction of Ori and range values were displayed in Table 1. For this study, a total of 17 experiments including five replications of the central points were designed. The model equation reflecting the effect of the three independent variables on the response (Y) was shown as follows in Equation (2):(2)$${Y=\beta }_{0}+\overset{3}{\underset{i=1}{\sum }}\beta_{i}X_{i}+\overset{3}{\underset{i=1}{\sum }}\beta_{\mathrm{ii}}X_{i}^{2}+\overset{2}{\underset{i=1}{\sum }}\overset{3}{\underset{j=2}{\sum }}\beta_{\mathrm{ji}}X_{i}X_{j}$$
where Y is the response variable, β_0_, β_i_, β_ii_, and β_ij_ are the regression coefficients for intercept, linearity, quadratic coefficients, and interaction coefficients, respectively, while X_i_ and X_j_ are the independent variables.

#### 2.2.3. Purification and Identification of Ori by PHPLC

The crude extract (1.0 g) was dissolved in methanol (25 mL), and then filtered through 0.45 μm microfiltration membrane (Millipore). The separation and purification of Ori were performed on an Agilent 1260 infinity PHPLC (Waldbronn, Germany) equipped with Agilent ZORBAX SB C_18_ column (9.4 mm × 250 mm, 5 μm particle size, Santa Rosa, CA, USA) using gradient elution. The flow rate, detection wavelength, and injection volume were 6 mL/min, 239 nm, and 1 mL, respectively. After repeated purification by PHPLC, the obtained Ori was identified by ^13^C-NMR, and the purity was determined by UPLC.

### 2.3. Preparation and Characterization of Ori Liposomes

#### 2.3.1. Preparation of Ori Liposomes

The Ori liposomes were prepared by film-dispersion and hydration-sonication methods. In brief, soybean phospholipids (160 mg), cholesterol (30 mg), Ori (20 mg), and chloroform (50 mL) were added to a round bottom flask, and dissolved by using ultrasonic dispersion technique to form an oil phase. Then, the organic solvent was removed by a rotary evaporator at 45 °C, and the lipid film was dried in a vacuum for 12 h. Afterwards, phosphate-buffered saline (PBS, pH 7.4, 10 mL) was added to hydrate for 15 min, and the mixturewas shaken and sonicated using a probe-type sonicator for 40 min under a 37 °C water bath. Finally, the liposome suspension was filtered using 0.45 μm membranes.

#### 2.3.2. Optimization of Ori Liposomes Preparation

According to the preliminary experiments, three variables at three levels (−1, 0, +1) were analyzed to study the impact on the encapsulation efficiency (EE) of Ori liposomes. The variables and their levels are displayed in Table 2. The ultrasonic time (min), soybean phospholipids/drug ratio (*w*/*w*), and the temperature of water bath (°C) were selected as independent variables, while EE was chosen as the response (Y). A three level-three variable BBD was used to optimize the preparation conditions of Ori liposomes using Design-Expert V12.0 software. The amount of the soybean phospholipids (160 mg), cholesterol (30 mg), and chloroform (50 mL) remained constant during the experiments. To minimize the influences of unexplained variability, the experiments were carried out in a randomized order.

#### 2.3.3. Characterization of Ori Liposomes

A slightly modified gel mini-column filtration method was used to determine the EE and drug loading (DL) of Ori liposomes in this study [25]. Sephadex G-50 was soaked in distilled water for 24 h, and then transferred to a 2 mL plastic syringe with a piece of filter paper placed at the bottom, and centrifuged to remove excessive water. The liposomes suspension was added to the column, and eluted with distilled water. The eluted Ori liposomes were collected and ruptured by methanol. The encapsulated amount of Ori in the liposomes was analyzed using UPLC (Waters ACQUITY UPLC)on a waters BEHC_18_ column (2.1 mm × 50 mm, 1.7 μm) at a wavelength of 239 nm, with mobile phase of acetonitrile-water (30:70, *v*/*v*) at a flow rate of 0.3 mL/min [24]. The EE and DL were calculated by the following Equations (3) and (4):EE (%) = W_en_/W_total_ × 100%(3)
DL (%) = W_en_/W_lipo_ × 100%(4)
where W_total_ was the total amount of Ori used in the preparation of liposomes, W_en_ was the amount of Ori encapsulated in liposomes, and W_lipo_ was the total weight of Ori liposomes.

The average particle size, polydispersity index (PDI), and zeta potential (Zeta) of the Ori liposomes were determined by means of the dynamic light scattering (DLS) technique using ZEN3690 Malvern Zetasizer Nano (Malvern instruments, Malvern, UK). 

### 2.4. Cytotoxity Test

The cytotoxicity of Ori solution, Ori liposomes, and blank liposomes on MCF-7 breast cancer cells was determined using MTT assay. MCF-7 cells in log phase of growth were seeded into a 96-well plate at a density of 8 × 10^3^ cells/well and then incubated at 37 °C and 5% CO_2_ for 24 h [26]. Different concentrations (10, 20, 30, 40, 50 μmol/L) of Ori liposomes and Ori solution were added. The cytotoxicity of blank vehicle was also subjected to the same procedure. After continuous culture for 24 h, 10 μL of MTT solution (5 mg/mL) were added to each well, and the incubation was terminated after 4 h. The culture medium was carefully removed, and 200 μL of DMSO was added to each well, and then the mixture was shaken for 10 min. The absorbance value (OD) of each well was determined at 490 nm with a BioTec Cytation 5 cell imaging multi-mode reader (Winooski, VT, USA). DMSO was used as a positive control, and blank liposomes were used as a negative control. The survival rate of cells was calculated according to the following Equation (5):Cell viability(%) = (OD_test_ − OD_positive control_)/(OD_negativecontrol_ − OD_positive control_) × 100%(5)

OD_test_ was the absorbance value of test compound; OD_positive control_ was the absorbance value of the positive control; and OD_negative control_ was the absorbance value of the negative control.

### 2.5. Cellular Uptake Study

Labeling liposomes with fluorescent dyes is considered a promising method to evaluate the cellular uptake. Coumarin 6 (C6), a derivative of coumarin, emits green fluorescence that can be used as a fluorescent probe to evaluate the cell uptake. Confocal laser scanning microcopy (CLSM) and fluorometric methods were used to assess intracellular uptake efficiency. MCF-7 cells were seeded in laser confocal dish at a density of 1 × 10^4^ cells/well and incubated at 37 °C for 24 h. Then, 20 μL of coumarin-6 and liposomes encapsulating coumarin-6 solutions in the concentration range of 10–50 μmol/mL were added. After 4 h incubation, the medium was removed and the cells were washed three times with cold phosphate buffer solution (PBS). Further, cells were fixed with 4% (*w*/*v*) chloral hydrate at room temperature and the cell nuclei were stained using DAPI (4′,6-diamidino-2-phenylindole). Blank liposomes were used as a negative control. Then, it was observed using a laser confocal microscope.

### 2.6. Statistical Analysis

All measurements were performed in triplicates, and the results were expressed as mean ± standard deviation (SD). The experimental data were analyzed using SPSS software 22.0. A value of *p* < 0.05 was considered significant.

## 3. Results and Discussion

### 3.1. Extraction and Purification of Ori from R. rubescens

#### 3.1.1. Single Factor Experimental Analysis

The effect of ethanol concentration on Ori yield is given in Figure 1a. As can be observed, the Ori yield increased gradually until an ethanol concentration of 60%, when the yield reached a peak and then declined. This phenomenon mainly can be attributed to a better solubility of Ori at higher ethanol concentrations. Tetracyclic diterpenoids are weak polarity compounds with low solubility. As the number of substituted hydroxyl groups increases, the polarity becomes stronger and the solubility increases accordingly. Ori has four hydroxyl groups, which increases its polarity, and has the best solubility in 60% ethanol. According to the results, ethanol concentration of 60% was the optimal concentration for UAE since it achieved the highest Ori extraction yield from *R. rubescens*.

According to the previous investigation, ultrasonic cavitation refers to the dynamic process of growth and collapse of micro-bubbles in the liquid that vibrate under the action of ultrasound when the ultrasound waves pressure reached a certain value [27,28]. It affected the surface of the plant matrix and promoted the release of internal ingredients. Generally, the longer the ultrasonic time was, the higher the extraction yield was; however, excessive ultrasonic time may cause the degradation of the bioactive compounds [29]. The effect of extraction time on yield of Ori from *R. rubescens* is presented in Figure 1b. The Ori yield increases sharply as extraction time increases from 10 min to 30 min, and then the further extension of the extraction time resulted in a decrease in Ori yield. Therefore, the extraction time of 30 min was chosen for UAE. Liquid/solid ratio is another important factor influencing the extraction yield. In this study, UAE process was performed using liquid/solid ratio in the range of 10:1 to 50:1, while ethanol concentration and ultrasonic time were fixed at 60% and 30 min, respectively. As shown in Figure 1c, the Ori yield increased rapidly with increasing liquid/solid ratio, and achieved a maximum at the liquid/solid ratio of 30:1, then there was no significant change as liquid/solid ratio continued to increase from 30:1 to 50:1. Therefore, liquid/solid ratio of 30:1 is suitable for the extraction process. 

Single factor experiments were performed to obtain the optimal extraction conditions by measuring their influences to Ori yield, in which one independent variable was changed while other factors remained constant. According to the results of single factor study, the following conditions were employed for the RSM experiments: ethanol concentration from 40% to 80%, extraction time of 30 to 50 min, and liquid/solid ratio of 20:1 to 40:1.

#### 3.1.2. Optimization of UAE by RSM 

According to the BBD design, 17 experiments were performed in triplicate and the obtained results are displayed in Table 3. Through multiple regression analysis of the experimental data, the model for the response variable and independent variables was expressed by the following quadratic polynomial Equation (6):(6)$$Y=-2.39748+0.08179X_{1}+0.16671X_{2}+0.02945X_{3}-0.00073X_{1}X_{2}+0.00083X_{1}X_{3}-0.00034X_{2}X_{3}-0.00053X_{1}^{2}-0.00147X_{2}^{2}-0.00109X_{3}^{2}$$

The analysis of variance (ANOVA) of response variable for the regression model is summarized in Appendix A. The model *F*-value was 33.54 and *p*-value was less than 0.01, meaning that the model was statistically significant. The determination coefficient (*R*^2^ = 0.9713) suggests that only 2.87% of the total variations are not explained by this model. The adjusted coefficient of determination (*R*^2^_a_ = 0.9344) is close to the determination coefficient, indicating that the model fitted well with the experimental data, revealing a good adequacy. The lack of fit value (*p* = 0.0968) was higher than 0.05 indicate that the model is appropriate for the description of experimental data. Moreover, the linear coefficients X_1_, all quadratic coefficients (X_1_^2^, X_2_^2^ and X_3_^2^) and two interaction coefficients (X_1 × 2_ and X_1_X_3_) showed significant effect on the extraction yield (*p* < 0.05). However, the effect of other regression coefficients, namely the linear coefficients X_2_ and X_3_, and the interaction coefficient X_2_X_3_, was insignificant (*p* > 0.05). The low coefficient variation (C.V. = 1.77%) value revealed a highly degree of precision and reliability of the experimental values of the regression model. 

#### 3.1.3. Analysis of Response Surfaces

The 3D response surface plots show the interactions between two continuous variables when the third variable is fixed at zero level. As can be seen from Figure 2a–c, all the response surfaces were a convex shape, indicating that the range of variables is selected appropriately. It can be seen from Figure 2a that the Ori yield increased by increasing ethanol concentration (X_1_) and extraction time (X_2_), but, with a longer extraction time of, the Ori yield decreased as ethanol concentration increased. Figure 2b showed the interaction between ethanol concentration (X_1_) and liquid/solid ratio (X_3_) on the Ori yield. Initially, the Ori yield increased rapidly by increasing ethanol concentration and liquid/solid ratio, but subsequently it decreased slowly. Figure 2c illustrated the simultaneous effect of extraction time and liquid/solid ratio on Ori yield. It can be seen that the Ori yield reached the maximum value at the extraction time of 35 min and liquid/solid ratio of 33:1 mL/g. More solvent would interfere with the ultrasonic process as the viscosity and surface tension of the solvent affect the cavitation effect, which makes the formation of bubbles difficult, thereby reducing the extraction efficiency [30].

#### 3.1.4. Verification of Predictive Model

By using the Design Expert 12.0 software, the optimal conditions for the extraction of Ori were obtained and recognized as ethanol concentration (X_1_) of 75.9%, extraction time (X_2_) of 35.7 min, and liquid/solid ratio (X_3_) of 32.6 mL/g. The predicted Y value, under these optimal conditions, was 4.28 mg/g. The experimental value was 4.23 ± 0.26 mg/g (*n* = 3) under the above modified conditions. No significant differences were found between the predicted and experimental results on Ori yield (*p* > 0.05). The result indicated that the model designed for the present study could be used for extraction of Ori from *R. rubescens*.

#### 3.1.5. Purity and Identification of Ori

After separation by PHPLC and purification by recrystallization, the purity of the isolated Ori was as high as 97.5% by UPLC analysis. The ^13^C-NMR spectra of Ori were obtained by a Bruker Avance 600 instrument using DMSO-d6 as solvent. The ^13^C-NMR data of Ori were as follows: δ: 209.0 (C-15), 152.5 (C-16), 119.8 (C-17), 97.4 (C-7), 73.6 (C-6), 72.9 (C-14), 72.1 (C-1), 63.1 (C-20), 62.0 (C-8), 59.9 (C-5), 53.5 (C-9), 43.2 (C-13), 40.9 (C-10), 38.8 (C-3), 33.8 (C-4), 33.2 (C-18), 30.5 (C-12), 29.8 (C-2), 22.2 (C-19), and 19.8(C-11). By comparison of the NMR spectral data with those reported in the literature [31], the results indicated that Ori has been successfully isolated.

### 3.2. Optimization of Liposomes Preparation by RSM

In this study, a 17-run BBD was designed to optimize the three independent variables, including X_1_ (ultrasonic time), X_2_ (soybean phospholipids/drug ratio), and X_3_ (temperature of water bath). The experimental design and the results of dependent variable (Y, EE) are displayed in Table 4.

After multiple regression analysis of various independents, the quadratic polynomial regression equation was obtained as follows in Equation (7):(7)$$Y=-154.5932+1.8436X_{1}+9.4868X_{2}+7.1395X_{3}-0.1394X_{1}X_{2}+0.00028X_{1}X_{3}+0.2075X_{2}X_{3}-0.01745X_{1}^{2}-2.6633X_{2}^{2}-0.07361X_{3}^{2}$$

The obtained regression model was tested by analysis of variance (ANOVA), in which the F-test and *p* value were used to describe the significance of the regression equation. The smaller *p* value was, the more significant the interaction among independent variables was. According to the ANOVA results of Table 5, the model *F*-value of 18.55 and *p* value < 0.01 indicated the model was significant, and the model could predict the actual experimental data. Meanwhile, the lack of fit (*p* = 0.0883) was not significant, which suggested that the theoretical values could be fitted to the experimental results. Additionally, the value of determination coefficient (*R*^2^ = 0.9598) and adjusted determination coefficient (*R*^2^_adj_ = 0.9098) also confirmed that the model fits well. Two independent variables (X_1_ and X_2_), and two quadratic terms (X_1_^2^ and X_3_^2^) significantly affected the EE of liposomes within a 99% confidence interval, and the independent variable X_3_, the interaction X_1_X_2_, and the quadratic term X_2_^2^ were significant (*p* < 0.05). However, the coefficient of interactions X_1_X_3_ and X_2_X_3_ were found to be non-significant (*p* > 0.05). 

After the 17-run BBD of preparation parameters, the optimum conditions for the preparation of Ori liposomes were obtained through regression model in accordance with the maximal yield of EE using Design expert software. The optimized conditions achieved by the 3D response surfaces (Figure 3) were an ultrasonic time of 41.1 min, a soybean phospholipids/drug ratio of 9.6 g/g, and a water bath temperature of 53.4 °C, under which the predicted EE was 84.34%. Three parallel verification trails were carried out under aforementioned optimum conditions, and the experimental value for EE was 84.1 ± 1.28%. The experimental results agreed closely with the predicted EE and consequently indicated that the RSM model is satisfactory and accurate.

### 3.3. Characterization of Ori Liposomes

As shown in Figure 4, the particle size of Ori liposomes (diameter) was distributed in the range of 100–300 nm, and the average particle size was 137.7 nm. The polydispersity index (PDI) was 0.216. For liposomal drug delivery system, a PDI value of 0.3 or below is considered to be acceptable, which indicates a homogeneous distribution of lipid vesicles [32]. The zeta-potentials of the Ori liposomes were about −24.0 mV, and the DL was determined to be 8.9%.

### 3.4. Cytotoxity Test

It can be seen from Figure 5 that the blank liposomes were not cytotoxic to MCF-7 cells, while the Ori solution and Ori liposomes had concentration-dependent inhibitory effects on MCF-7 breast cancer cells. The IC_50_ of Ori liposomes was 49.4 ± 3.65 μmol/L, and the free Ori was 86.1 ± 7.32 μmol/L. The difference between the free Ori and liposomes was statistically significant (*p* < 0.01). This might be due to the high affinity of liposomes to the cell membrane, and Ori that was encapsulated in the liposomes, which can enter the cells more easily and exert better anti-tumor effect [33].

### 3.5. Cell Uptake Assay

DAPI is a widely used nuclear counterstain for fluorescent techniques. The blue fluorescent in Figure 6a showed that nuclei were stained with DAPI. Coumarin 6 is a fluorescent probe widely used for staining polymer particles, organelles, or other materials used in medicine. The green fluorescence within the cytoplasm indicating that the liposomes were taken up by MCF-7 cells (Figure 6a). As shown in Figure 6b, the fluorescence intensity of C6 liposomes was significantly higher than that of free C6. The main reason was that coumarin 6 entered cells through passive diffusion while liposomes came into cells via endocytosis [34,35]. Endocytosis promoted liposomes entry and accumulation in specific cells [36].

## 4. Conclusions

The application of RSM by BBD was employed to study the effect of UAE conditions on the extraction Ori from *R. rubescens*. The high correlation of the models demonstrated that the second-order polynomial model could be successfully used for optimizing the extraction parameters. The results showed that ethanol concentration influenced the Ori yield markedly. The optimized conditions of UAE were ethanol concentration of 75.9%, extraction time of 35.7 min, and liquid/solid ratio of 32.6 mL/g, which resulted in 4.23 ± 0.26 mg/g of Ori yield. Ori liposomes were successfully prepared by film-dispersion and hydration-sonication methods, and the optimal preparation conditions for Ori liposomes were obtained by RSM, corresponding to an ultrasonic time of 41.1 min, a soybean phospholipids/drug ratio of 9.6 g/g, and a water bath temperature of 53.4 °C. The EE of Ori liposomes was 84.1 ± 1.28%, and the Ori liposomes exhibited reasonable values of particle size distribution, zeta potential, and PDI. Results of in vitro cell uptake studies indicated the cumulative uptake of Ori liposomes, which was significantly higher than Ori solution. All these findings demonstrated that Ori liposomes could enhance the anti-tumor activity of Ori and had the potential to act as an efficient anti-breast cancer drug in vivo.

## Figures and Tables

**Figure 1 molecules-27-00860-f001:**
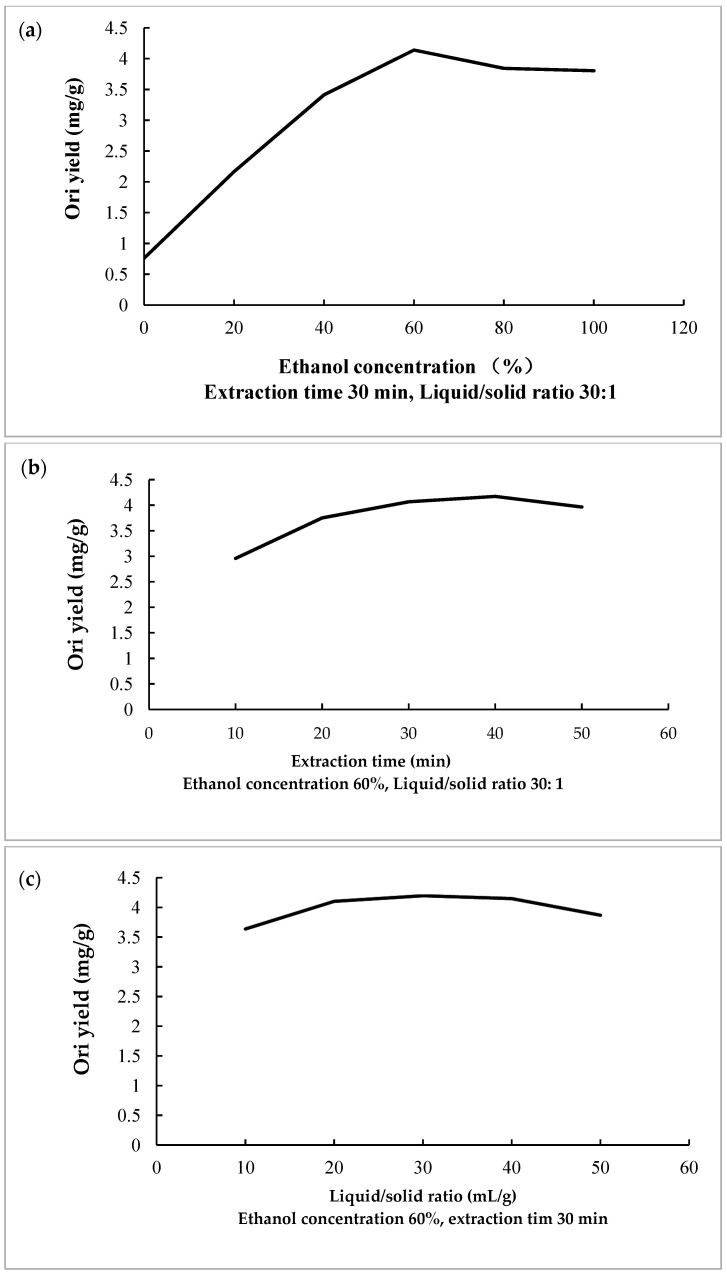
The influence of different factors on the extraction yield of Ori. (**a**) Effect of ethanol concentration on the yield of Ori, (**b**) Effect of extraction time on the yield of Ori, (**c**) Effect of liquid/solid ratio on the yield of Ori.

**Figure 2 molecules-27-00860-f002:**
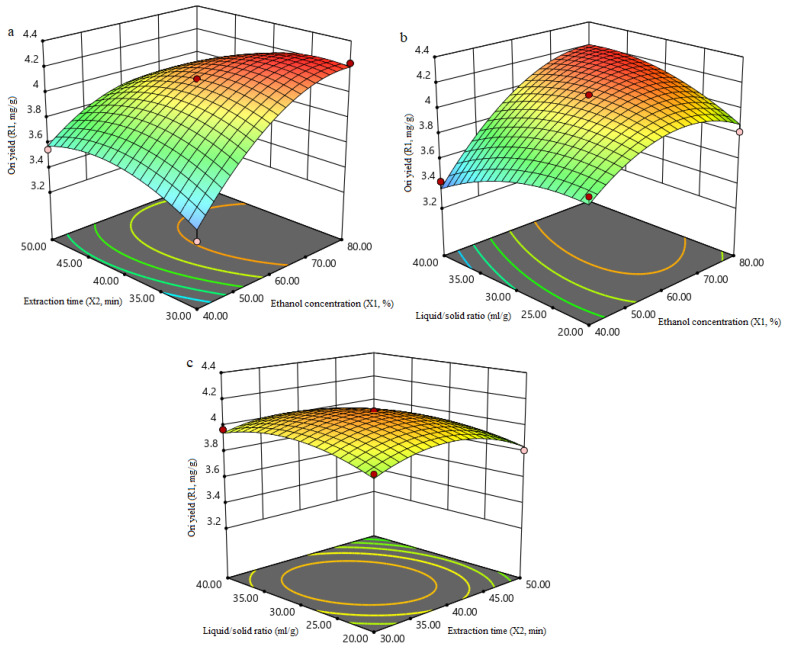
3D response surface plots of independent variables in extraction yield of Ori: (**a**) Effects of ethanol concentration and extraction time on the yield of Ori; (**b**) Effects of ethanol concentration and Liquid/solid ratio on the yield of Ori; (**c**) Effects of etraction time and Liquid/solid ratio on the yield of Ori.

**Figure 3 molecules-27-00860-f003:**
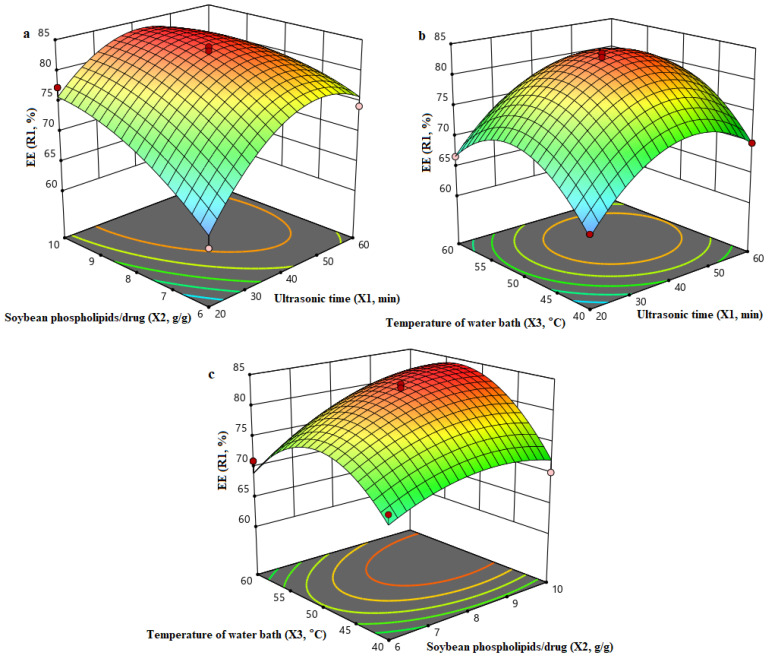
3D response surface of independent variables in EE of Ori liposomes: (**a**) Effects of ultrasonic time and soybean phospholipids/drug on the EE; (**b**) Effects of soybean phospholipids/drug and temperature of water bath on the EE; (**c**) Effects of soybean phospholipids/drug and temperature of water bath on the EE.

**Figure 4 molecules-27-00860-f004:**
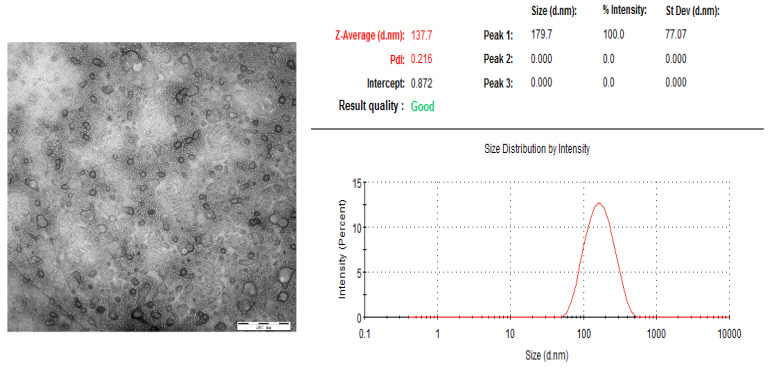
Transmission Electron Microscope (TEM) image and size distribution of Ori-liposomes.

**Figure 5 molecules-27-00860-f005:**
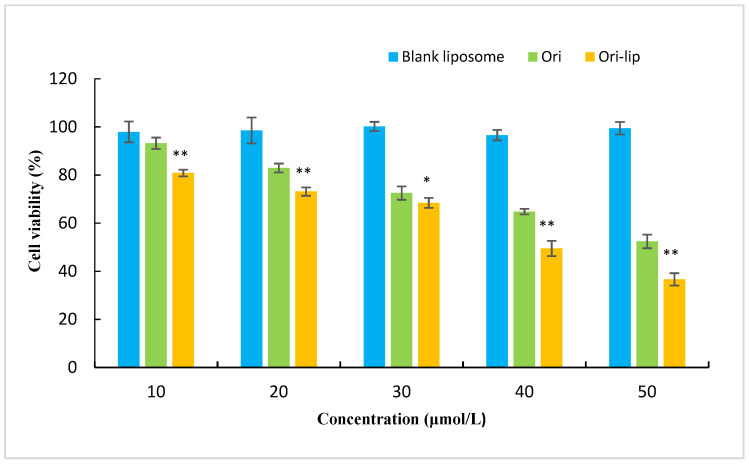
Cell survival curves after treating MCF-7cells with different concentrations of blank liposomes, free Ori, and Ori liposomes for 24 h. ** p* < 0.05, ** *p* < 0.01.

**Figure 6 molecules-27-00860-f006:**
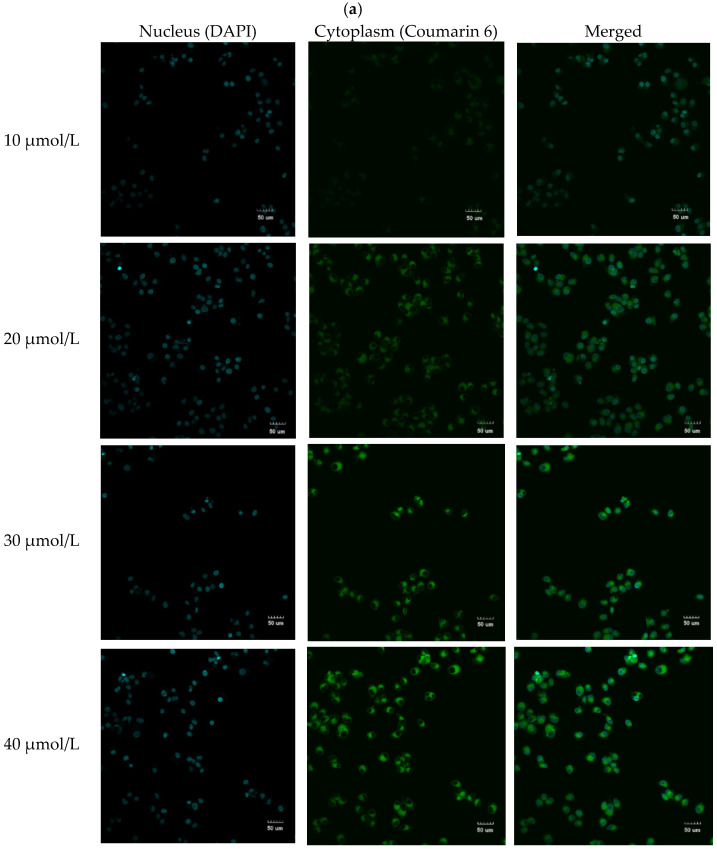
Cellular uptake of C6 and C6-liposomes into MCF-7 cells: (**a**) Microscopy images of MCF-7 cells treated with different concentrations C6-liposomes for 4 h; (**b**) Fluorescent intensity of C6 and C6-liposomes in MCF-7 cells after incubation for 4 h. ** *p* < 0.01.

**Table 1 molecules-27-00860-t001:** Levels of factors for the UAE.

Independent Variables	Code	−1	0	1
Ethanol concentration (%)	X_1_	40	60	80
Extraction time (min)	X_2_	30	40	50
Liquid/solid ratio (mL/g)	X_3_	20:1	30:1	40:1

**Table 2 molecules-27-00860-t002:** Levels of independent variables for the Ori liposomes preparation.

Independent Variables	Code	−1	0	1
Ultrasonic time (min)	X_1_	20	40	60
Soybean phospholipids/drug ratio (g/g)	X_2_	6:1	8:1	10:1
Temperature of water bath (°C)	X_3_	40	50	60

**Table 3 molecules-27-00860-t003:** BBD matrix and response values for the extraction yield of Ori.

	Ethanol Concentration	Extraction Time (min)	Liquid/Solid Ratio (mL/g)	Ori Yield (mg/g)	
				Experimental	Predicted
1	0	−1	1	3.96	3.94
2	0	0	0	4.11	4.11
3	−1	1	0	3.56	3.58
4	0	0	0	4.11	4.11
5	−1	0	1	3.42	3.36
6	−1	−1	0	3.31	3.4
7	1	1	0	3.89	3.8
8	0	1	−1	3.81	3.83
9	0	1	1	3.73	3.76
10	0	−1	−1	3.90	3.87
11	0	0	0	4.11	4.11
12	1	−1	0	4.23	4.21
13	0	0	0	4.11	4.11
14	1	0	1	4.15	4.21
15	−1	0	−1	3.75	3.7
16	1	0	−1	3.82	3.88
17	0	0	0	4.11	4.11

**Table 4 molecules-27-00860-t004:** BBD design and experimental encapsulation efficiency (%).

	Ultrasonic Time	Soybean Phospholipids/Drug	Temperature of Water Bath	EE (%)	
				Experimental	Predicted
1	0	0	0	83.89	82.66
2	−1	0	−1	63.14	62.67
3	0	1	−1	70.13	72.15
4	1	0	1	73.62	74.09
5	0	0	0	81.92	82.66
6	1	1	0	79.52	77.53
7	−1	−1	0	60.95	62.94
8	−1	0	1	66.63	66.66
9	1	0	−1	69.91	69.88
10	0	0	0	83.67	82.66
11	0	−1	1	71	68.98
12	0	−1	−1	70.56	69.03
13	−1	1	0	77.34	75.78
14	0	0	0	80.61	82.66
15	0	0	0	83.23	82.66
16	1	−1	0	74.28	75.84
17	0	1	1	78.87	80.4

**Table 5 molecules-27-00860-t005:** Results of ANOVA analysis.

Effects	Source	Sum of Squares	Degree of Freedom (DF)	Mean Square	*F* Value	*p*-Value
	model	802.47	9	89.16	18.55	0.0004 ^a^
Linear	X_1_	107.09	1	107.09	22.28	0.0022 ^a^
	X_2_	105.63		105.63	21.97	0.0022 ^a^
	X_3_	33.54		33.54	6.98	0.0334 ^a^
Interaction	X_1_X_2_	31.08		31.08	6.47	0.0385 ^a^
	X_1_X_3_	0.0121		0.0121	0.0025	0.9614 ^b^
	X_2_X_3_	17.22		17.22	3.58	0.1003 ^b^
Quadratic	X_1_^2^	205.04		205.04	42.65	0.0003 ^a^
	X_2_^2^	29.86		29.86	6.21	0.0414 ^a^
	X_3_^2^	228.13	1	228.13	47.46	0.0002 ^a^
	Residual	33.65	7	4.81		
	Lack of fit	26.04	3	8.68	4.56	0.0883 ^b^
	Pure error	7.61	4	1.90		
	Cor. total	836.12	16			

*R*^2^ = 0.9598, adjusted *R*^2^ = 0.9098 C.V.% = 2.94%. ^a^ 5% significance level. ^b^ Not significant relative to the pure error.

## Data Availability

Not applicable.

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
