# Peer review of "Preparation, Characterization, and Evaluation of Liposomes Containing Oridonin from *Rabdosia rubescens"

_molecules, 2022, doi:10.3390/molecules27030860_

Round 1
Reviewer 1 Report
The document “preparation, characterization, and evaluation of liposomes containing oridonin from Rabdosia rubescens”, is an interesting manuscript, well described with well performed experiments, some minimum comments need to be addressed prior publication.
In the material and methods section please indicate if the UAE equipment uses a bath or a probe and indicate the temperature control system. For the UPLC conditions it is necessary to add a reference or the conditions for R. rubescens determination and quantification.
In the results and discussion section in paragraph from line 177 to 192 please indicate how the temperature was controlled, and if not indicate its influence in the process. Also explain chemically how a better solubilization is achieved of Ori at 60%.
In figure 1 please indicated the factor that is fixed in each graphic, i.e. in graphic “a” indicate the extraction time and liquid ratio. Add error bars in each graphic.
Table 4 can be added as supplementary data.
Author Response
Dear reviewer,
Thank you for your comments concerning our manuscript. We appreciate the reviewer’s positive evaluation of our work. Those comments are valuable and very helpful. We have read comments carefully and have made corrections. Based on the instructions provided in your letter, we uploaded the file of the revised manuscript.
1 In the material and methods section please indicate if the UAE equipment uses a bath or a probe and indicate the temperature control system. For the UPLC conditions it is necessary to add a reference or the conditions for R. rubescens determination and quantification.
Response:We are grateful for the suggestion. As suggested by the reviewer, we have added more details of the UAE equipment (water bath and temperature control panel) and the reference of UPLC conditions.
2 In the results and discussion section in paragraph from line 177 to 192 please indicate how the temperature was controlled, and if not indicate its influence in the process. Also explain chemically how a better solubilization is achieved of Ori at 60%.
Response:Thank you for your comments. UAE was carried out in a 30 °C water bath (Line 92). We have explained the reason from chemical structure in the revised manuscript (line 199 to 203).
3 In figure 1 please indicated the factor that is fixed in each graphic, i.e. in graphic “a” indicate the extraction time and liquid ratio. Add error bars in each graphic.
Response:We are extremely grateful to reviewer for pointing out this problem. We have added the details in Figure 1.
4 Table 4 can be added as supplementary data.
Response:Thank you for your comments. Table 4 was used as supplementary data.
Reviewer 2 Report
The manuscript entitled “Preparation, characterization, and evaluation of liposomes containing oridonin from Rabdosia rubescens” intends to optimize the ultrasound assisted extraction (UAE) of oridonin (Ori) and its further encapsulation into liposomes aiming a final application as a cancer drug delivery system. Even though a few previous studies focused on the encapsulation of Ori into liposomes, optimization studies of the Ori extraction by an eco-friendly technology and encapsulation into liposomes were performed for the first time in this research. The manuscript is well-structured. Although the abstract presents the objectives and main results, it should be summarized since it is too long. The introduction provides sufficient background and clearly presents the objectives. However, the relevance and novelty of the study should be carefully addressed in the introduction along with the aims of the study. The experimental design seems accurate and well executed, but some information is missing in the methods section regarding controls and blanks used. The results were properly presented together with the statistical analyses. However, the discussion lacks comparisons and reasoning based on the literature. Possible explanations for the results obtained and their relevance in this research field is also missing and should be provided to obtain a more critical analysis of the outcomes. The conclusion evidences the main results and conclusions of the study. The paper may also benefit from the inclusion of a few more references. Some questions are addressed below and should be answered before the publication of manuscript in Molecules journal:
- In introduction (lines 36-37), the authors stated that “Ori effectively inhibit the growth and proliferation in a variety of human cells”. What did the authors refer with “inhibit the growth and proliferation”? Of malignant cells? Of bacteria? Please revise it.
- Some information needs to be added or clarified in the methods section, including the preparation of liposomes and amounts of each ingredient, as well as optimization of processes by RSM. For instance, in subsection 2.3.1, the authors stated “the certain amount of lecithin…" (line 107). What does this mean? The authors should indicate the amounts used of each ingredient/excipient. Please revise this subsection carefully.
- In addition, in subsection 2.4 from methods’ section, the authors need to include information about the control and blank used and in what concentrations if applied (lines 142-154). Also, usually in this assay, the terminology used is positive control and negative control instead of blank and control. Please revise it.
- In subsection 2.5 (lines 156-163), the authors should also explain why they used coumarin-6 liposomes. What was the purpose of this assay? Was a blank or control (positive or negative control) used? Please revise it carefully and provide the information required.
- Why did the authors perform a single factor analysis prior to the RSM (lines 171-197)? Please explain it. This explanation should be also provided in the discussion section as well as the relevance of this analysis for the study.
Other minor comments:
- Lines 5-7: Revise the affiliations’ presentation. The e-mails of each author followed by the name initials of authors within parentheses are missing.
- Line 9: The authors stated “Rabdosia rubescens has attracted more and more attention”. In what field? Pharmaceutical field? Please revise it.
- Line 13: Box-Behnken design (BBD) is one experimental design method of Response surface methodology (RSM) as well as CCD (central composite design). In this way, please revise this sentence by replacing "and" for "namely" (“Response surface methodology (RSM), namely Box-Behnken design (BBD), was applied…”).
- Lines 15-16: Consider deleting this sentence.
- Line 19: Delete the parentheses and the standard deviation, rewriting it as “was 23 mg/g". Also, delete “by the model”.
- Line 21: Add a space between number and unit (“53.4 °C”).
- Line 22: Is “entrapment efficiency” or encapsulation efficiency? Please consider indicating “1%” within parentheses and delete “of” (“efficiency (84.1%).”). Correct “was” to “were” (“Ori liposomes were”).
- Line 25: Consider revising the sentence as “…indicated that liposomes could enhance the bioactivity of Ori, being proposed as a promising vehicle...”.
- Line 27: Replace the semicolon “;” by a point.
- Line 33: Format “ent” in italic (“ent-kaurane”).
- Line 34: Add a space before the references (“ rubescens [1, 2].”). Also add, a space before “According”. Revise all the manuscript for these minor mistakes.
- Lines 35-36: Delete “the” and revise this part of the sentence (as suggestion, “anti-inflammatory, scavenging activity against oxygen free radicals as well as antibacterial”).
- Line 37: Correct “inhibits”.
- Line 39: Correct “limited” to “limits”.
- Lines 42-43: Use “vesicles” (plural) and “were”.
- Line 44: Revise this part of the sentence (as suggestion, "...allows the encapsulation of polar drug molecules.").
- Line 48: Define the abbreviation “FDA”.
- Line 50: Delete “the” (“for anti-cancer…”).
- Line 54: Use “responses” (plural).
- Line 58: Define the abbreviation “BBD”.
- Lines 63-65: The sentence seems a little confusing. Please consider revising it (as suggestion, “Based on the optimal conditions, physicochemical characteristics of the Ori liposomes as well as anti-proliferation activity of breast cancer cells (MCF-7) were investigated.").
- Line 66: Consider using another verb to replace “observe” (for example, appraise, evaluate, determine or assess).
- Line 67: Replace “provide” for “provides”.
- Lines 72-73: Revise the letter format (style and size) throughout the manuscript. Some parts are unformatted.
- Line 73: Add “a” (“in a desiccator”).
- Line 79: Delete “procedures” from the subtitle of the section 2.2.1.
- Line 80: Rewrite “powdered sample”.
- Line 81: Revise “ethanol concentration” as “ethanol at different concentrations”. Also, rewrite “in a flask” by replacing “the” for “a”.
- Line 82: Add a space between number and unit (“100 W”).
- Line 83: Revise the sentence (“and the oridonin content was determined”).
- Line 84: Revise the sentence as “as follows in equation (1):”. Also, in the equation, consider replacing “powder weight (g)” by “powdered sample weight (g)”.
- Line 85: Use “design” (singular) instead of “designs” in the subtitle of section 2.2.2.
- Lines 86-87: Revise the sentence as “BBD was employed through RSM to optimize…” by using only the abbreviations BBD and RSM since the full names are already defined above.
- Line 90: Correct the word “displayed”.
- Line 91: Consider replacing “trails” by “experiments”.
- Line 92: Add “the” (“of the three independent…”).
- Line 93: Revise the sentence as “as follows in equation (2):”. Delete the parentheses in the beginning of equation (2) before “Y”.
- Lines 90-93: The authors should state that the response studied for these independent variables was oridonin extraction yield. Please add this information in this subsection.
- Table 1: Revise the table by replacing “Figure 1” for “independent variables”.
- Lines 99 and 100: “Angilient” or Agilent? Add a space between number and unit “9.4 mm”.
- Line 104: Add “and” in the subtitle of section 2.3 (“2.3. Preparation and characterization…”).
- Line 106: Replace “was” by “were”.
- Line 108: Correct the word “bottom”.
- Line 109: Add a space between number and unit “45 °C”. Revise all the manuscript for this mistake.
- Line 110: Replace “And then” by “Afterwards”.
- Line 119: Replace “factors” by “independent variables”.
- Line 120: Correct the word “Design”.
- Line 122: Please consider rewriting “in a randomized order”.
- Table 2: Revise the subtitle (as suggestion, "Levels of independent variables for the..."). In the table, replace “factors” by “independent variables” and indicate the units of ultrasonic time and temperature within parentheses (“ultrasonic time (min)” and “temperature of water bath (ºC)”).
- Line 131: Correct “was” to “were”.
- Line 133: Add “a” (“at a wavelength”).
- Line 135: Start the sentence with capital letter "T.." and correct “by the following equations (3) and (4):”.
- Line 137: Add “the” (“the amount of Ori encapsulated”) and correct the word “liposomes”.
- Line 145: Correct “concentrations” (plural).
- Line 147: Replace “way” by “procedure”.
- Line 151: Indicate the microplate reader used within parentheses.
- Line 162: Indicate the full name of “DAPI”.
- Line 166: Use “mean” (singular) instead of “means”.
- Line 170: Revise the subtitle of section 3.1 by adding “and” instead of comma (“3.1. Extraction and purification…”).
- Line 172: Add a space between “Figure 1a”.
- Line 173: Replace “that” by a comma.
- Line 174: Add “and” (“reached a peak and then it declined”). Add “a” (“to a better”).
- Line 175: Use “concentrations” (plural) (“at higher ethanol concentrations”).
- Lines 175-176: “appropriate” or optimal concentration? Also, please indicate clearly that 60% was the optimal ethanol concentration since it achieved the highest Ori extraction yield from rubescens.
- Line 177: Rewrite “refers” instead of “referred”.
- Line 179: “sound” or ultrasound waves?
- Line 185: Correct “decrease” (instead of “decreased”).
- Line 186: Consider using “liquid/solid ratio” instead of “liquid/material ratio”. Revise it in all manuscript.
- Line 189: Add a space between “Figure 1c”.
- Figure 1: In figure 1a, please consider rewriting “Ethanol concentration (%)” in the axis x. In figure 1b, correct the unit to “min” in the axis x. In figure 1c, revise the identification of the axis x to “Liquid/solid ratio (mL/g)”.
- Lines 205-206: Revise the sentence as “was expressed…equation (6):”. In equation (6), delete the parentheses in the beginning of the equation before “Y”.
- Line 213: Revise the sentence (as suggestion, “…the model fitted well with the experimental data, revealing a good adequacy.”)
- Lines 215-216: Revise the sentence as "the linear coefficient X1, all quadratic coefficients (...) and two interaction coefficients (...)".
- Lines 217-218: Please indicate the coefficients that showed no significance ("…regression coefficients, namely the linear coefficients X2 and X3, and the interaction coefficient X2X3, was insignificant (P>0.05).")
- Table 4: In the subtitle, add “ANOVA” within parentheses after “Analysis of variance”. In table, name the first column as "Effects" and delete the word "effects" after "linear", "interaction" and "quadratic" placed in the lines below. Define the abbreviation “DF” and format "p" form “p-value” in italic letter. In the footnotes (line 223), format “R” from “R2” in italic letter.
- Line 226: Delete the point at the end of the subtitle 3.1.3.
- Line 228: Use the word “variable” (singular) (“the third variable”). Correct to “Figure 2 (a-c)” Also, correct “Fig.” to “Figure” in lines 230, 232 and 235.
- Line 229: Delete “graph”.
- Line 236: Correct “to” to “on” (“ratio on Ori yield.”).
- Lines 236-237: Please consider revising the sentence as “Ori yield reached the maximum value at the...”.
- Line 238: Correct “ultrasonic”.
- Line 239: Correct “affecting” to “affect”.
- Line 240: Place the point after the reference.
- Figure 2: To be easier to the readers to identify the variables on each response surface, please define them properly in the axes. For example, in this Figure, "R1 (mg/g)" should be rewritten as "Ori yield (R1, mg/g)". The X1 should be “Ethanol concentration (X1, %)”, while X2 should be “Extraction time (X2, min)” and X3 should be “liquid/solid ratio (X3, mL/g)”. Please consider revising it carefully. In the subtitle, consider adding “independent” before “variables”. In addition, in the subtitle, include the identification of each graph identified as “a”, “b” and “c”.
- Line 247: Please delete the parentheses and rewrite the result as “4.23±0.26 mg/g”. Also, format “n” in italic letter (“(n=3)”).
- Line 248: What did the authors mean with "was consistent"? Does that mean there was not significant differences between the predicted and experimental results on Ori yield since p<0.05? Please explain it and revise it in the manuscript.
- Lines 249-250: Rewrite this part of the sentence by adding “"...extraction of oridonin from rubescens.".
- Line 251: Consider revising the subtitle of section 3.1.5 as “3.1.5. Purity and identification of oridonin” and delete “determination”.
- Line 257: Add “and” (“(C-19), and8(C-11).”
- Line 260: Also, please consider revising the subtitle of section 3.2 “3.2. Optimization of liposomes preparation by RSM”.
- Table 6: In the subtitle, format “Results of ANOVA analysis” in non-italic letter. Similar to Table 4, name the first column as "Effects" and delete the word "effects" after "linear", "interaction" and "quadratic" placed in the lines below. Define the abbreviation “DF” and format "p" form “p-value” in italic letter. In the footnotes (line 223), format “R” from “R2” in italic letter.
- Line 271: Rewrite “as follows in equation (7):”.
- Line 274: Add “variables” after “independent”.
- Line 278: Instead of “calculated values” use “theoretical values” or “predicted values”. Also, rewrite “fitted to” instead of “fit with”.
- Lines 279-280: The “R” from the determination coefficient “R2“ should be formatted in italic letter. Revise all the manuscript for this mistake.
- Lines 280-283: The sentence is confusing. Please revise the sentence particularly for the use of "while" used to make a contrast between two ideas. However, all the independent variables and interactions mentioned in the sentence revealed to be significant (p<0.05), so the ideas mentioned are similar and not contrary. Also, revise the sentence for some minor grammatical and spelling mistakes. For example, in lines 280-281, delete “The” and correct “independent” and “significantly” (“ Two independent variables (X1 and X2), and two quadratic terms (X12 and X32) significantly affected …”). Consider rewriting in line 282 as “while the independent variable X3, the interaction X1X2 and the quadratic term X22 were”.
- Line 283: Add “interactions” (“coefficient of interactions X1X3 and X2X3”).
- Figure 3: Identify the axes properly similar to the above Figure 2. In the subtitle, add “independent” before “variables”. In addition, in the subtitle, include the identification of each graph identified as “a”, “b” and “c”.
- Lines 287-288: Add “the” and “of” (“for the preparation of Ori liposomes”).
- Line 290: Use “surfaces” (plural).
- Line 292: Correct “is” to “was”.
- Line 293: Delete the parentheses and rewrite it as “84.1±1.28%”.
- Line 294: Correct “agreeing” to “agreed” and “indicating” to “indicated”.
- Line 297: Correct “were” to “was”.
- Line 300: Place the point after the reference.
- Line 302: In the subtitle of Figure 4, define “TEM” using the full name.
- Line 304: Rewrite “were” instead of “was”.
- Lines 306-307: Rewrite both results without parentheses as “49.4±3.65” and “86.1±7.32”.
- Line 308: Reformat “P” in italic letter (“(P<0.01)”).
- Line 310: Place the point after the reference. Do the same in line 319.
- Figure 5: Format the letter of graph in black color. Please consider revising the subtitle (as suggestion, “…different concentrations of blank liposomes, free Ori and Ori liposomes for 24 h.”).
- Figure 6: In the subtitle, rewrite “C6-liposomes” instead of “C6-lip”. Also, add a space between number and unit (“4 h”). Is it "**P<0.01" or with three as "***P<0.01"? In the subtitle is only mentioned “**”, but in the figure it is mentioned “***”.
- Line 323: Rewrite “by” instead of “couple”. Also, correct "Box–Behnken”.
- Line 324: Rewrite “UAE conditions on the extraction of Ori…”.
- Line 326: Rewrite “for optimizing the extraction parameters”.
- Line 327: Please consider revising as “The optimized conditions of UAE were ethanol…”.
- Line 329: Rewrite the result without the parentheses as “23±0.26 mg/g”. Do the same in line 333 (“84.1±1.28%”). Replace “Ori sample” by “Ori yield”. Correct “was” to “were”.
- Line 331: Rewrite “corresponding to” instead of “as follows”.
- Line 335: Add “which” before “was”.
- Lines 336-337: Add “of Ori” (“anti-tumor activity of Ori”). Add the word “drug” as “anti-breast cancer drug in vivo”.
- Line 342: Revise all the references for minor mistakes (for example, in line 345, format “in vivo” and “in vitro” in italic letter; and in line 346, format “ent” from “ent-kaurane” in italic letter).
Author Response
Dear reviewer,
Thank you for your precious comments and advice. Those comments are all valuable and very helpful for revising and improving our paper, as well as the important guiding significance to our researches. We have studied comments carefully and have made correction which we hope meet with approval.
As suggested by reviewer, we have summarized the abstract (delete 13-14, 17-18). And we have described the novelty of this study in the introduction (Line 68-72). Moreover, the control and the blank group were also added in the revised manuscript (Line 170-172, 188).
- In introduction (lines 36-37), the authors stated that “Ori effectively inhibit the growth and proliferation in a variety of human cells”. What did the authors refer with “inhibit the growth and proliferation”? Of malignant cells? Of bacteria? Please revise it.
Response:Thank you for your comment. It refers the human cancer cells (Line 41)
- Some information needs to be added or clarified in the methods section, including the preparation of liposomes and amounts of each ingredient, as well as optimization of processes by RSM. For instance, in subsection 2.3.1, the authors stated “the certain amount of lecithin…" (Line 107). What does this mean? The authors should indicate the amounts used of each ingredient/excipient. Please revise this subsection carefully.
Response:Thank you for your suggestion. As suggested by reviewer, we have added the suggested content to the manuscript on page 3 (Line 122,123, Line 137-139).
- In addition, in subsection 2.4 from methods’ section, the authors need to include information about the control and blank used and in what concentrations if applied (lines 142-154). Also, usually in this assay, the terminology used is positive control and negative control instead of blank and control. Please revise it.
Response:We have added the positive and negative control in revised manuscript (Line 170-176).
- In subsection 2.5 (lines 156-163), the authors should also explain why they used coumarin-6 liposomes. What was the purpose of this assay? Was a blank or control (positive or negative control) used? Please revise it carefully and provide the information required.
Response:We deeply appreciate the reviewer’s suggestion. According to the reviewer’s comment, we have added a more detailed interpretation regarding the coumarin-6 (Line 178-180, 374-346)
- Why did the authors perform a single factor analysis prior to the RSM (line 171-197)? Please explain it. This explanation should be also provided in the discussion section as well as the relevance of this analysis for the study.
Response:We have described it in revised manuscript (Line 226-228).
Other minor comments:
- Lines 5-7: Revise the affiliations’ presentation. The e-mails of each author followed by the name initials of authors within parentheses are missing.
- Response:We have added the E-mails of each author (Line 5-8).
- Line 9: The authors stated “Rabdosia rubescens has attracted more and more attention”. In what field? Pharmaceutical field? Please revise it.
- Response:We have added “pharmaceutical field” (Line 10).
- Line 13: Box-Behnken design (BBD) is one experimental design method of Response surface methodology (RSM) as well as CCD (central composite design). In this way, please revise this sentence by replacing "and" for "namely" (“Response surface methodology (RSM), namely Box-Behnken design (BBD), was applied…”).
- Response:Thank you for your comment. We have revised it.
- Lines 15-16: Consider deleting this sentence.
- Response:We have deleted this sentence in revised manuscript.
- Line 19: Delete the parentheses and the standard deviation, rewriting it as “was 23 mg/g". Also, delete “by the model”.
- Response:We have deleted the parentheses, the standard deviation and “by the model”.
- Line 21: Add a space between number and unit (“53.4 °C”).
- Response:We have added a space.
- Line 22: Is “entrapment efficiency” or encapsulation efficiency? Please consider indicating “1%” within parentheses and delete “of” (“efficiency (84.1%).”). Correct “was” to “were” (“Ori liposomes were”).
- Response:We are very sorry for our negligence. Encapsulation efficiency is right. “of” was deleted, and “was” to “were” (line 24, 25).
- Line 25: Consider revising the sentence as “…indicated that liposomes could enhance the bioactivity of Ori, being proposed as a promising vehicle...”.
- Response:Thank you for your suggestion. We have revised as you suggested (Line 28).
- Line 27: Replace the semicolon “;” by a point.
- Response:We have replace the semicolon “;” by a point.
- Line 33: Format “ent” in italic (“ent-kaurane”).
- Response:We have modified it.
- Line 34: Add a space before the references (“rubescens [1, 2].”). Also add, a space before “According”. Revise all the manuscript for these minor mistakes.
- Response: We have modified these mistakes throughout the text.
- Lines 35-36: Delete “the” and revise this part of the sentence (as suggestion, “anti-inflammatory, scavenging activity against oxygen free radicals as well as antibacterial”).
- Response: We have revised as you suggested (Line 39,39).
- Line 37: Correct “inhibits”.
- Response: We are very sorry for our negligence. We have modified this mistake.
- Line 39: Correct “limited” to “limits”.
- Response: We are very sorry for our negligence. We have modified this mistake.
- Lines 42-43: Use “vesicles” (plural) and “were”.
- Response: We have modified this mistake.
- Line 44: Revise this part of the sentence (as suggestion, "...allows the encapsulation of polar drug molecules.").
- Response:Thank you for your suggestion. We have revised as you suggested (Line 48).
- Line 48: Define the abbreviation “FDA”.
- Response:We have added the full name of “FDA”(Line 53).
- Line 50: Delete “the” (“for anti-cancer…”).
- Response:We have deleted “the”.
- Line 54: Use “responses” (plural).
- Response: We have modified this mistake.
- Line 58: Define the abbreviation “BBD”.
- Response:We have added the full name of “BBD” (Line 64).
- Lines 63-65: The sentence seems a little confusing. Please consider revising it (as suggestion, “Based on the optimal conditions, physicochemical characteristics of the Ori liposomes as well as anti-proliferation activity of breast cancer cells (MCF-7) were investigated.").
- Response:Thank you for your suggestion. We have revised as you suggested (Line 71,72).
- Line 66: Consider using another verb to replace “observe” (for example, appraise, evaluate, determine or assess).
- Response:Thank you for your suggestion. The verb of assess was used in revised manuscript.
- Line 67: Replace “provide” for “provides”.
- Response: We are very sorry for our negligence. We have c
- Lines 72-73: Revise the letter format (style and size) throughout the manuscript. Some parts are unformatted.
- Response: We have revised the letter format throughout the manuscript.
- Line 73: Add “a” (“in a desiccator”).
- Response: We have modified this mistake.
- Line 79: Delete “procedures” from the subtitle of the section 2.2.1.
- Response: We have modified this mistake.
- Line 80: Rewrite “powdered sample”.
- Response: We have modified this mistake.
- Line 81: Revise “ethanol concentration” as “ethanol at different concentrations”. Also, rewrite “in a flask” by replacing “the” for “a”.
- Response: We are very sorry for our negligence. We have modified these mistakes.
- Line 82: Add a space between number and unit (“100 W”).
- Response: We have add a space between number and unit throughout the manuscript.
- Line 83: Revise the sentence (“and the oridonin content was determined”).
- Response:Thank you for your suggestion. We have revised as you suggested (Line 93,94).
- Line 84: Revise the sentence as “as follows in equation (1):”. Also, in the equation, consider replacing “powder weight (g)” by “powdered sample weight (g)”.
- Response:Thank you for your suggestion. We have revised as you suggested.
- Line 85: Use “design” (singular) instead of “designs” in the subtitle of section 2.2.2.
- Response: We are very sorry for our negligence. We have modified this mistake.
- Lines 86-87: Revise the sentence as “BBD was employed through RSM to optimize…” by using only the abbreviations BBD and RSM since the full names are already defined above.
- Response:Thank you for your suggestion. We have revised as you suggested (Line 98).
- Line 90: Correct the word “displayed”.
- Response: We are very sorry for our incorrect writing.
- Line 91: Consider replacing “trails” by “experiments”.
- Response: We have replaced the “trails” by “experiments”.
- Line 92: Add “the” (“of the three independent…”).
- Response: We have added the “the” in this sentence.
- Line 93: Revise the sentence as “as follows in equation (2):”. Delete the parentheses in the beginning of equation (2) before “Y”.
- Response: We have modified these mistakes.
- Lines 90-93: The authors should state that the response studied for these independent variables was oridonin extraction yield. Please add this information in this subsection.
- Response: We have added these information in this sussection.
- Table 1: Revise the table by replacing “Figure 1” for “independent variables”.
- Response: We have modified this mistake.
- Lines 99 and 100: “Angilient” or Agilent? Add a space between number and unit “9.4 mm”.
- Response: We have modified this mistake and added a space.
- Line 104: Add “and” in the subtitle of section 2.3 (“2.3. Preparation and characterization…”).
- Response: We have modified this mistake.
- Line 106: Replace “was” by “were”.
- Response: We have modified this mistake.
- Line 108: Correct the word “bottom”.
- Response: We have modified this mistake.
- Line 109: Add a space between number and unit “45 °C”. Revise all the manuscript for this mistake.
- Response: We have modified these mistakes.
- Line 110: Replace “And then” by “Afterwards”.
- Response: We have replaced “And then” by “Afterwards”.
- Line 119: Replace “factors” by “independent variables”.
- Response: We have replaced “factors” by “independent variables”.
- Line 120: Correct the word “Design”.
- Response: We have modified this mistake.
- Line 122: Please consider rewriting “in a randomized order”.
- Response:Thank you for your suggestion. We have revised as you suggested.
- Table 2: Revise the subtitle (as suggestion, "Levels of independent variables for the..."). In the table, replace “factors” by “independent variables” and indicate the units of ultrasonic time and temperature within parentheses (“ultrasonic time (min)” and “temperature of water bath (ºC)”).
- Response:Thank you for your suggestion. We have revised as you suggested.
- Line 131: Correct “was” to “were”.
- Response: We have modified this mistake.
- Line 133: Add “a” (“at a wavelength”).
- Response: We have modified this mistake.
- Line 135: Start the sentence with capital letter "T.." and correct “by the following equations (3) and (4):”.
- Response: We have modified this mistake.
- Line 137: Add “the” (“the amount of Ori encapsulated”) and correct the word “liposomes”.
- Response: We have modified thess mistakes.
- Line 145: Correct “concentrations” (plural).
- Response: We have modified this mistake.
- Line 147: Replace “way” by “procedure”.
- Response: We have replaced the “way” by “procedure”.
- Line 151: Indicate the microplate reader used within parentheses.
Response:We have deleted the parentheses and added the details of the microplate reader.
- Line 162: Indicate the full name of “DAPI”.
- Response: We have added the full name of “DAPI”.
- Line 166: Use “mean” (singular) instead of “means”.
- Response: We have modified this mistake.
- Line 170: Revise the subtitle of section 3.1 by adding “and” instead of comma (“3.1. Extraction and purification…”).
- Response:Thank you for your suggestion. We have revised as you suggested.
- Line 172: Add a space between “Figure 1a”.
- Response: We have modified this mistake.
- Line 173: Replace “that” by a comma.
- Response: We have modified this mistake.
- Line 174: Add “and” (“reached a peak and then it declined”). Add “a” (“to a better”).
- Response: We have modified this mistake.
- Line 175: Use “concentrations” (plural) (“at higher ethanol concentrations”).
- Response: We have modified this mistake.
- Lines 175-176: “appropriate” or optimal concentration? Also, please indicate clearly that 60% was the optimal ethanol concentration since it achieved the highest Ori extraction yield fromrubescens.
- Response:Thank you for your suggestion. It is optimal concentration. We have revised as you suggested.
- Line 177: Rewrite “refers” instead of “referred”.
- Response: We have modified this mistake.
- Line 179: “sound” or ultrasound waves?
- Response:Thank you for your suggestion. We have modified as ultrasound wave.
- Line 185: Correct “decrease” (instead of “decreased”).
- Response: We have modified this mistake.
- Line 186: Consider using “liquid/solid ratio” instead of “liquid/material ratio”. Revise it in all manuscript.
- Response:We have replaced material by solid.
- Line 189: Add a space between “Figure 1c”.
- Response: We are very sorry for our negligence.
- Figure 1: In figure 1a, please consider rewriting “Ethanol concentration (%)” in the axis x. In figure 1b, correct the unit to “min” in the axis x. In figure 1c, revise the identification of the axis x to “Liquid/solid ratio (mL/g)”.
- Response:Thank you for your suggestion. We have revised as you suggested.
- Lines 205-206: Revise the sentence as “was expressed…equation (6):”. In equation (6), delete the parentheses in the beginning of the equation before “Y”.
- Response:Thank you for your suggestion. We have revised as you suggested.
- Line 213: Revise the sentence (as suggestion, “…the model fitted well with the experimental data, revealing a good adequacy.”)
- Response:Thank you for your suggestion. We have revised as you suggested (Line 251,252).
- Lines 215-216: Revise the sentence as "the linear coefficient X1, all quadratic coefficients (...) and two interaction coefficients (...)".
- Response:Thank you for your suggestion. We have revised as you suggested (Line 256, 257).
- Lines 217-218: Please indicate the coefficients that showed no significance ("…regression coefficients, namely the linear coefficients X2 and X3, and the interaction coefficient X2X3, was insignificant (P>0.05).")
- Response:Thank you for your suggestion. We have revised as you suggested (Line 256, 257).
- Table 4: In the subtitle, add “ANOVA” within parentheses after “Analysis of variance”. In table, name the first column as "Effects" and delete the word "effects" after "linear", "interaction" and "quadratic" placed in the lines below. Define the abbreviation “DF” and format "p" form “p-value” in italic letter. In the footnotes (line 223), format “R” from “R2” in italic letter.
- Response:Thank you for your suggestion. We have revised as you suggested.
- Line 226: Delete the point at the end of the subtitle 3.1.3.
- Response: We have deleted the point.
- Line 228: Use the word “variable” (singular) (“the third variable”). Correct to “Figure 2 (a-c)” Also, correct “Fig.” to “Figure” in lines 230, 232 and 235.
- Response: We have modified these mistakes.
- Line 229: Delete “graph”.
- Response: We have deleted the “graph”.
- Line 236: Correct “to” to “on” (“ratio on Ori yield.”).
- Response: We have corrected “to” to “on”.
- Lines 236-237: Please consider revising the sentence as “Ori yield reached the maximum value at the...”.
- Response:Thank you for your suggestion. We have revised as you suggested.
- Line 238: Correct “ultrasonic”.
- Response: We are very sorry for our incorrect writing. We have modified this mistake.
- Line 239: Correct “affecting” to “affect”.
- Response: We have corrected “affecting” to “affect”.
- Line 240: Place the point after the reference.
- Figure 2: To be easier to the readers to identify the variables on each response surface, please define them properly in the axes. For example, in this Figure, "R1 (mg/g)" should be rewritten as "Ori yield (R1, mg/g)". The X1 should be “Ethanol concentration (X1, %)”, while X2 should be “Extraction time (X2, min)” and X3 should be “liquid/solid ratio (X3, mL/g)”. Please consider revising it carefully. In the subtitle, consider adding “independent” before “variables”. In addition, in the subtitle, include the identification of each graph identified as “a”, “b” and “c”.
- Response:Thank you for your suggestion. We have revised the Figure 1 as you suggested.
- Line 247: Please delete the parentheses and rewrite the result as “4.23±0.26 mg/g”. Also, format “n” in italic letter (“(n=3)”).
- Response: The parentheses were deleted and “n” is in italic.
- Line 248: What did the authors mean with "was consistent"? Does that mean there was not significant differences between the predicted and experimental results on Ori yield since p<0.05? Please explain it and revise it in the manuscript.
- Response:Thank you for your suggestion. We have revised this sentence as you suggested.
- Lines 249-250: Rewrite this part of the sentence by adding “"...extraction of oridonin fromrubescens.".
- Response:Thank you for your suggestion. We have revised as you suggested
- Line 251: Consider revising the subtitle of section 3.1.5 as “3.1.5. Purity and identification of oridonin” and delete “determination”.
- Response:Thank you for your suggestion. We have revised as you suggested
- Line 257: Add “and” (“(C-19), and8(C-11).”
- Response:We have added the “and”.
- Line 260: Also, please consider revising the subtitle of section 3.2 “3.2. Optimization of liposomes preparation by RSM”.
- Response:Thank you for your suggestion. We have revised as you suggested.
- Table 6: In the subtitle, format “Results of ANOVA analysis” in non-italic letter. Similar to Table 4, name the first column as "Effects" and delete the word "effects" after "linear", "interaction" and "quadratic" placed in the lines below. Define the abbreviation “DF” and format "p" form “p-value” in italic letter. In the footnotes (line 223), format “R” from “R2” in italic letter.
- Line 271: Rewrite “as follows in equation (7):”.
- Response: We have revised as you suggested.
- Line 274: Add “variables” after “independent”.
- Response:We have added “variables” after “independent”.
- Line 278: Instead of “calculated values” use “theoretical values” or “predicted values”. Also, rewrite “fitted to” instead of “fit with”.
- Response:Thank you for your suggestion. We have revised as you suggested
- Lines 279-280: The “R” from the determination coefficient “R2“ should be formatted in italic letter. Revise all the manuscript for this mistake.
- Response: We have modified this expression throughout the text according to your comment.
- Lines 280-283: The sentence is confusing. Please revise the sentence particularly for the use of "while" used to make a contrast between two ideas. However, all the independent variables and interactions mentioned in the sentence revealed to be significant (p<0.05), so the ideas mentioned are similar and not contrary. Also, revise the sentence for some minor grammatical and spelling mistakes. For example, in lines 280-281, delete “The” and correct “independent” and “significantly” (“ Two independent variables (X1 and X2), and two quadratic terms (X12 andX32) significantly affected …”). Consider rewriting in line 282 as “while the independent variable X3, the interaction X1X2 and the quadratic term X22 were”.
- Response: We are grateful for the suggestion. We have corrected it.
- Line 283: Add “interactions” (“coefficient of interactions X1X3 and X2X3”).
- Response: We have added this word.
- Figure 3: Identify the axes properly similar to the above Figure 2. In the subtitle, add “independent” before “variables”. In addition, in the subtitle, include the identification of each graph identified as “a”, “b” and “c”.
- Response: The subtitle was corrected.
- Lines 287-288: Add “the” and “of” (“for the preparation of Ori liposomes”).
- Response: We have added “the” and “of”.
- Line 290: Use “surfaces” (plural).
- Response: We have modified this mistake.
- Line 292: Correct “is” to “was”.
- Response: We have modified this mistake.
- Line 293: Delete the parentheses and rewrite it as “84.1±1.28%”.
- Response: We have modified this mistake.
- Line 294: Correct “agreeing” to “agreed” and “indicating” to “indicated”.
- Response: We have modified this mistake.
- Line 297: Correct “were” to “was”.
- Response: We have modified this mistake.
- Line 300: Place the point after the reference.
- Response: We have modified this mistake.
- Line 302: In the subtitle of Figure 4, define “TEM” using the full name.
- Response: We have added the full name (Line 361).
- Line 304: Rewrite “were” instead of “was”.
- Response: We have modified this mistake.
- Lines 306-307: Rewrite both results without parentheses as “49.4±3.65” and “86.1±7.32”.
- Response: We have deleted the parentheses.
- Line 308: Reformat “P” in italic letter (“(P<0.01)”).
- Response: We have modified “P” in italic letter.
- Line 310: Place the point after the reference. Do the same in line 319.
- Response: We have modified this mistake.
- Figure 5: Format the letter of graph in black color. Please consider revising the subtitle (as suggestion, “…different concentrations of blank liposomes, free Ori and Ori liposomes for 24 h.”).
- Response:Thank you for your suggestion. We have revised as you suggested (line 371,372).
- Figure 6: In the subtitle, rewrite “C6-liposomes” instead of “C6-lip”. Also, add a space between number and unit (“4 h”). Is it "**P<0.01" or with three as "***P<0.01"? In the subtitle is only mentioned “**”, but in the figure it is mentioned “***”.
- Response:Thank you for your suggestion. We have revised as you suggested (line 381-383).
- Line 323: Rewrite “by” instead of “couple”. Also, correct "Box–Behnken”.
- Response:Thank you for your suggestion. We have revised as you suggested
- Line 324: Rewrite “UAE conditions on the extraction of Ori…”.
- Line 326: Rewrite “for optimizing the extraction parameters”.
- Response:Thank you for your suggestion. We have revised as you suggested
- Line 327: Please consider revising as “The optimized conditions of UAE were ethanol…”.
- Response:Thank you for your suggestion. We have revised as you suggested.
- Line 329: Rewrite the result without the parentheses as “23±0.26 mg/g”. Do the same in line 333 (“84.1±1.28%”). Replace “Ori sample” by “Ori yield”. Correct “was” to “were”.
- Response: We have deleted the parentheses.
- Line 331: Rewrite “corresponding to” instead of “as follows”.
- Response: We have replaced the “as follows” to “corresponding to”.
- Line 335: Add “which” before “was”.
- Response: We have added the “which” before “was”.
- Lines 336-337: Add “of Ori” (“anti-tumor activity of Ori”). Add the word “drug” as “anti-breast cancer drug in vivo”.
- Response: We have added the words of “of Ori” and “drug”.
- Line 342: Revise all the references for minor mistakes (for example, in line 345, format “in vivo” and “in vitro” in italic letter; and in line 346, format “ent” from “ent-kaurane” in italic letter).
- Response: We have modified this expression throughout the text according to your comment.
Thank you for your careful review. We really appreciate your efforts in reviewing our manuscript during this unprecedented and challenging time. We wish good health to you, your family, and community. Your careful review has helped to make our study clearer and more comprehensive.
Reviewer 3 Report
Dear authors,
Please consider the suggestions below:
2.2.3
L99 - Agilent? Include the series of HPLC, the series. city and country
L100 – include brand, city and country of column manufacturer. How the sample was prepared for analysis? Include in the text, the solvent the sample was dissolved, and specifications of filter. Was the sample cleaned with SPE? If yes, include.
Topic 2.4
Include the equipment used to measure absorbance (brand/model/city/country)
Topic 3
L181-182 – excessive US time may degrade the bioactive compounds.
Topic 3.1.1
L184 /fig1b – Is the extraction time in min or h? Correct this.
Tables 3/5/- include the statistical differences within the treatments
Figure 6 -The micrographs before the graphic identified in Fig 6 was not identified. Write in the text the effects of samples in cells
Author Response
Thank you for your comments concerning our manuscript. Those comments are all valuable and very helpful for revising and improving our paper, as well as the important guiding significance to our researches. We have studied comments carefully and have made correction which we hope meet with approval.
2.2.3 L99 - Agilent? Include the series of HPLC, the series. city and country, L100 – include brand, city and country of column manufacturer.
Response:We are very sorry for our incorrect writing. And we have added the details about the instruments.
How the sample was prepared for analysis? Include in the text, the solvent the sample was dissolved, and specifications of filter. Was the sample cleaned with SPE? If yes, include.
Response:We have described the sample preparation in detail (Line111-115).
Topic 2.4 Include the equipment used to measure absorbance (brand/model/city/country)
Response:We have added the details of the microplate reader (Line 169).
Topic 3 L181-182 – excessive US time may degrade the bioactive compounds.
Response:
Topic 3.1.1
L184 /fig1b – Is the extraction time in min or h? Correct this.
Response:We are very sorry for our negligence. Extraction time is in min. We have corrected it.
Tables 3/5/- include the statistical differences within the treatments
Response:We have added it in the Table 3/5.
Figure 6 -The micrographs before the graphic identified in Fig 6 was not identified. Write in the text the effects of samples in cells
Response:We have re-written this part according to your suggestion (Line 372-374).
Once again, thank you very much for your comments and suggestions.
Round 2
Reviewer 2 Report
The authors revised carefully the manuscript according to the comments and suggestions provided by the reviewers. All the changes required were done, as well as the questions were properly answered. Additionally, I suggest some minor changes before considering the manuscript for publication in Molecules journal.
- In the title, please add a space between "Rabdosia" and "rubescens".
- Please revise carefully the manuscript formatting, using the same letter (style, size and space between lines) as well as eliminating the empty spaces.
- Line 34 (page 1): Add "properties" after "anti-bacterial".
- Line 37 (page 1): Delete the extra space after "limits".
- Line 42 (page 1): Add a space between "liposomes" and "allows" and correct the letter formatting in this sentence.
- Line 173 (page 2): Add a space between "methodology" and "(RSM)".
- Line 183 (page 2): Delete the extra space after "liposomes".
- Line 185 (page 2): Please add a space between "(MCF-7)" and "were".
- Line 261 (page 3): Add a space between "5 μm".
- Table 3 (page 9): Format the subtitle of table 3. Also, instead of "Actual value", consider rewriting it as "Experimental value".
- Lines 532-533 (page 12): If it is stated that there was no significant differences so p must be higher than 0.05 (p>0.05) right? Please revise it carefully.
- Line 549 (page 13): Add a space between "literature" and "[31]".
- Table 4 (page 13): Similar to Table 3, please consider rewriting "Actual value" as "Experimental value".
- Line 579 (page 14): Add a space between "variable" and "X3".
- Lines 631-632 (page 16): Add a space "49.4±3.65" and "µmol/l", as well as "86.1±7.32" and "µmol/l".
- Line 635 (page 16): Add a space between "effect" and "[33]".
- Lines 652-653 (page 17): Please add at least one reference to support this statement.
- Figure 6 (page 18): Reformat the subtitle of Figure 6.
- Line 668 (page 18): Use the abbreviation "BBD" defined above instead of the full name "Box-Behnken design".
- Line 672 (page 18): Delete the extra space between "were" and "ethanol".
- Line 678 (page 18): Add a space between "was" and "84.1±1.28%".
Author Response
- In the title, please add a space between "Rabdosia" and "rubescens".
- Response: We have added a space between "Rabdosia" and "rubescens" in revised manuscript.
- Please revise carefully the manuscript formatting, using the same letter (style, size and space between lines) as well as eliminating the empty spaces.
- Response: We have revised the manuscript formatting.
- Line 34 (page 1): Add "properties" after "anti-bacterial".
- Response: We have added "properties" after "anti-bacterial".
- Line 37 (page 1): Delete the extra space after "limits".
- Response: We have deleted the extra space after "limits".
- Line 42 (page 1): Add a space between "liposomes" and "allows" and correct the letter formatting in this sentence.
- Response: We have added the space and revised the formatting.
- Line 173 (page 2): Add a space between "methodology" and "(RSM)".
- Response: We have added the space.
- Line 183 (page 2): Delete the extra space after "liposomes".
- Response: We have deleted the extra space.
- Line 185 (page 2): Please add a space between "(MCF-7)" and "were".
- Response: We have added the space between "(MCF-7)" and "were"..
- Line 261 (page 3): Add a space between "5 μm".
- Response: We have added the space.
- Table 3 (page 9): Format the subtitle of table 3. Also, instead of "Actual value", consider rewriting it as "Experimental value".
- Response: We have replaced "Actual value" by "Experimental value".
- Lines 532-533 (page 12): If it is stated that there was no significant differences so pmust be higher than 0.05 (p>0.05) right? Please revise it carefully.
- Response: We are very sorry for our negligence. No significant differences were found between the predicted and experimental results on Ori yield (p>0.05).
- Line 549 (page 13): Add a space between "literature" and "[31]".
- Response: We have added the space.
- Table 4 (page 13): Similar to Table 3, please consider rewriting "Actual value" as "Experimental value".
- Response: We have replaced "Actual value" by "Experimental value".
- Line 579 (page 14): Add a space between "variable" and "X3".
- Response: We have added the space
- Lines 631-632 (page 16): Add a space "49.4±3.65" and "µmol/l", as well as "86.1±7.32" and "µmol/l".
- Response: We have added the space
- Line 635 (page 16): Add a space between "effect" and "[33]".
- Response: We have added the space between "effect" and "[33]".
- Lines 652-653 (page 17): Please add at least one reference to support this statement.
- Response: Two references were added in revised manuscript.
- Figure 6 (page 18): Reformat the subtitle of Figure 6.
- Response: The subtitle of Figure 6 was modified in revised manuscript.
- Line 668 (page 18): Use the abbreviation "BBD" defined above instead of the full name "Box-Behnken design".
- Response: "Box-Behnken design" was replaced by BBD.
- Line 672 (page 18): Delete the extra space between "were" and "ethanol".
- Response: We have deleted the extra space between "were" and "ethanol".
- Line 678 (page 18): Add a space between "was" and "84.1±1.28%".
- Response: We have added the space between "was" and "84.1±1.28%".
Thank you for your comments concerning our manuscript. Those comments are all valuable and very helpful for revising and improving our paper, as well as the important guiding significance to our researches. We have studied comments carefully and have made correction which we hope meet with approval.